# Taking a Closer Look at Teacher Support and Children’s Mental Health: The Mediating Role of Self-Concept and the Moderating Role of Area

**DOI:** 10.3390/bs15070987

**Published:** 2025-07-21

**Authors:** Zhongqi Chen, Qingqing Du, Bram Orobio de Castro, Guoxiong Liu

**Affiliations:** 1Department of Psychology, Nanjing Normal University, Nanjing 210097, China; chenzhongqi2021@outlook.com; 2Research Institute of Child Development and Education, Faculty of Social and Behavioural Sciences, University of Amsterdam, 1001 NG Amsterdam, The Netherlands; q.du@uva.nl

**Keywords:** perceived teacher support, self-concept, depression, anxiety, rural and urban area

## Abstract

Ample evidence has recognized the importance of teacher support for children’s mental health. However, less is known about the factors that may play a role in the association between teacher support and children’s mental health. In the present study, we examined the mediating role of self-concept and the moderating role of area (i.e., rural vs. urban area) in the relationship between teacher support and children’s mental health (i.e., depression and anxiety). We used a sample of Chinese elementary students who enrolled in grades 3 to 6 (*N* = 525, aged 8–13, 44.54% girls). Results showed that children who perceived more teacher support tended to report better mental health. Moreover, the relationship between perceived teacher support and mental health was mediated by each domain of self-concept. Lastly, the association between teacher support and mental health was moderated by area, with a stronger effect found for urban children compared with rural children. These findings highlight the importance of considering both individual and contextual factors in the association between teacher support and children’s mental health. Practical strategies are proposed for school teachers, professionals, and policymakers to foster children’s psychological well-being.

## 1. Introduction

A recent psychiatric epidemiological survey has revealed concerning trends in Chinese children’s mental health ([36]). Based on a large national sample (*N* = 17,524) and one-on-one clinical screening, the survey indicated that approximately 17.5% of Chinese children aged 6 to 16 were diagnosed with one or more mental disorders. This rate is higher than the global estimate of one in seven children ([69]). Evidence also suggested that prevalence had risen over the past three decades ([71]). Among these mental disorders, depression and anxiety are becoming increasingly prevalent, even among younger children, and may lead to further emotional and behavior problems, functional impairment, and even suicide ([8]; [55]; [66]).

Addressing these challenges requires a better understanding of factors that can mitigate psychological risks and promote mental health of children. Theoretically, teacher support, an important source of social support, not only helps children navigate adversity but also promotes growth and development in its absence ([16]; [26]). Consequently, teacher support appears to be important for children’s mental health ([27]; [57]; [58]). This has been supported by a wealth of empirical studies. However, most of research has been conducted in Western countries, leaving this association less well understood in Eastern contexts ([65]).

Although the link between teacher support and children’s mental health has been well established, the factors that may explain this relationship remain seem to be largely unknown. One of such factor is self-concept, referring to an individual’s perception of themselves. Perceived teacher support has been found to positively be associated with scholastic self-concept and to negatively predict behavior conduct problems ([48]). Moreover, self-concept has been found to be an important predictor of children’s mental health ([52]; [51]; [59]). That is, children with high self-concept are more likely to experience positive feelings and have better mental health ([31]). Despite these links, research has rarely examined the potential mediating role of self-concept in the association between teacher support and children’s mental health.

In addition to self-concept, area (rural versus urban) may be another factor that is relevant for the association between teacher support and mental health within cultures ([14]; [40]), especially considering substantial economic and social differences between urban and rural areas like China ([39]). These disparities may moderate the association between teacher support and mental health by influencing the accessibility and quality of support for children’s mental health. For instance, urban regions with more resources may amplify the benefits of support, whereas rural areas with limited resources may weaken this association. Thus far, virtually no research has studied the potential moderating role of area in this link.

To address these research gaps, the present study aimed to explore the association between teacher support and mental health in an Eastern context. Specifically, we investigated the potential mediating role of multidimensional self-concept and the moderating role of area (i.e., rural versus urban) in this association among Chinese children.

### 1.1. The Association Between Teacher Support and Children’s Mental Health

Theoretical perspectives have suggested that teacher support plays a crucial role in fostering children’s mental health. According to Social Support Theory, the support received from significant others such as teachers buffers the negative impact of stress and enhances coping abilities ([16]; [56]). Conversely, children who perceive lower levels of teacher support are more likely to experience anxiety and depression ([7]). Extending this perspective, the Thriving Through Relationships framework highlights that social support may not only protect against adversity but also facilitate personal growth in its absence ([26]). That is, in difficult times, teacher support may serve as a critical source of resilience and strength, whereas in the absence of adversity, it may as a catalyst for personal development and active engagement in meaningful life experiences.

Empirically, a considerable body of empirical research has supported the significance of teacher support for children’s mental health ([17]; [48]; [58]; [60]). Adolescents who perceived more teacher support were more likely to experience less depression one year later ([17]). Teacher support and the quality of the student–teacher relationship are inextricably linked. Children who experienced more stable, higher-quality teacher–child relationships in early childhood (aged 4–5) appeared to exhibit fewer mental health problems, including reduced emotional problems and hyperactivity, and stronger prosocial behaviors at aged 6 ([48]). A similar link between teacher support and mental health was found in elementary school as well. Specifically, teachers’ emotional warmth and investment of resources negatively predicts depressive symptoms over time, indicating that children who receive higher levels of teacher support are less likely to develop depressive symptoms in later years ([60]). Moreover, positive teacher–student relationship consistently predicts less anxiety over time ([58]). In summary, these studies collectively demonstrate that teacher support serves as a critical protective factor for children’s mental health. However, these finding are mainly based on research with Western populations, leaving unanswered whether the association between teacher support and mental health can be found in Eastern contexts.

### 1.2. Self-Concept as a Potential Mediator in the Association Between Teacher Support and Mental Health

In addition to the direct link between teacher support and mental health, factors such as self-concept may play a mediating role in this association. Self-concept refers to an individual’s perception of themselves, encompassing self-evaluations across multiple domains, such as scholastic competence, social and athletic skills, physical appearance, behavioral conduct, and global self-worth ([6]; [14]; [31]; [46]). As suggested by the Thriving Through Relationships framework, apart from direct effect of teacher support on mental health, teacher support may contribute to children’s mental health through psychological factors, such as positive self-evaluation. Specifically, when students receive support from teachers during difficult times, they may feel accepted and understood, and are likely to have a higher level of self-concept, thereby contributing to better mental health ([26]).

Empirically, teacher support has been found to be important for children’s self-concept across different domains. For instance, support from teachers has been found to shape children’s self-perceptions on scholastic competence ([23]; [44]). Children with closer teacher–student relationships and teacher support in sixth grade were more likely to have higher scholastic self-concept when they were in seventh grade ([19]). In addition to scholastic competence, teacher support has been shown to be relevant for children’s self-concept in behavioral conduct and global self-worth. Greater emotional teacher support has been found to be linked with lower children’s aggression ([47]), which is a key indicator of behavioral conduct. In a twelve-year longitudinal study, children’s behavioral conduct decreased concurrently with the increase of warmth in teacher–student relationship. In contrast, a decrease earlier in warmth concurrently appeared with an increase in behavioral conduct later ([24]). Moreover, children who had close relationship with their teachers earlier tended to have higher global self-worth later ([19]).

In addition to the link between teacher support and self-concept, numerous empirical studies have supported that high self-concept benefits children’s mental health (e.g., [3]; for a review; [30]; [51]). On the contrary, children with low self-concept tend to have social maladjustment, leading to more depression and anxiety (e.g., [52]; [50]; [59], for a meta-analytic review). For instance, a two-year longitudinal study found that general self-concept at age 10 negatively predicted depression at age 12, even after controlling for social support. However, the reverse predictive relationship was not supported ([53]). Another longer-term longitudinal study demonstrated that adolescents (aged 16–19) with higher self-concept in domains of global self-worth and social acceptance were less likely to use antidepressant in adulthood (aged 30) ([70]).

Even though the links between teacher support and self-concept and between self-concept and mental health were well-established, the mediating role of self-concept between teacher support and mental health seems to be rarely investigated. Thus far, only one study has investigated the relationships among the three variables ([9]). This study demonstrated indirect relationships through self-esteem between depression and family and peer support, but not for teacher support. Notably, this study was conducted among adolescents. It remains uncertain whether these findings can be applied to middle-to-late childhood (aged 8–12), a critical period for the development and expression of self-concept in children ([31]; [67]). Furthermore, this study investigated only a single, unidimensional measure of self-esteem without considering differentiating other important aspects of self-concept, such as self-perceived physical appearance, scholastic, athletic, and social competence, and behavioral conduct. These domains represent important components of self-concept during middle to late childhood ([31]; [64]), each of which may exert effects on the association between teacher support and mental health.

### 1.3. The Potential Moderating Role of Area in the Association Between Teacher Support and Mental Health

The role of teacher support in children’s mental health is particularly important when considering the substantial disparities between rural and urban area. As demonstrated in the case of China, the distribution of the socioeconomic structure in rural areas is characterized by a “pyramid-shaped” graph, whereas in urban areas, it is represented as an “olive-shaped” graph ([39]). Additionally, rural children are only one-tenth as likely to attend elite universities as their urban peers ([37]). Given these disparities, guaranteeing adequate teacher support for children in rural areas remains a significant challenge. For example, a follow-up study found that the turnover rate of school teachers in rural China was 12.06% within one year ([10]). In another survey with a larger sample (*N* = 7730), 65.7% of rural teachers hoped to move to urban areas to teach ([54]). In fact, teachers in rural elementary schools typically manage a class of 60 or more students, while class sizes in urban areas are often half as large ([74]). These structural disadvantages hinder the formation of supportive teacher–student relationships in rural area, making it difficult for teachers to satisfy individual students’ needs.

Although lower levels of teacher support are linked to poorer mental health in children, the strength of this association appears to differ across the context where children are. Some studies have investigated both left-behind—those whose parents, one or both, migrate to cities for work and do not live with them in rural areas—and non-left-behind children, but yield inconsistent findings. On the one hand, [40] ([40]) found that the positive effect of teacher support on mental health was less pronounced in left-behind children than in non-left-behind children. On the other hand, [43] ([43]) found that children who perceived higher-quality teacher–student relationship reported lower depression, with a stronger protective effect among left-behind children than among non-left-behind peers. Moreover, higher levels of teacher support was associated with lower anxiety among both left-behind and non-left-behind children, and mitigated anxiety by strengthening its association with attachment security only among left-behind children ([78]). Taken together, findings on the effect of teacher support on mental health between left-behind and non-left-behind children were inconsistent. To our best knowledge, so far, no study directly investigates how rural versus urban settings may exert an effect on the association between teacher support and mental health.

### 1.4. The Present Study

The present study aimed to advance our understanding of the association between teacher support and mental health (i.e., depression and anxiety) among Chinese elementary school children (grade 3 to 6). Specifically, we tested a conceptual model (see Figure 1) and investigated the mediating role of self-concept and the moderating role of area in this association.

Our research departed from prior research in at least three ways: First, it would contribute to the Thriving Through Relationships framework by providing empirical evidence from an underrepresented population—Chinese children. Second, these findings would demonstrate how teacher support related to children’s mental health by identifying multidimensional self-concept as a mediator and area as a moderator, thus contributing to the Thriving Through Relationships framework. Lastly, the study would provide practical implications by informing mental health interventions for children in disadvantaged contexts.

We formulated two research questions with corresponding hypotheses. First, we investigated the potential mediating role of self-concept in the association between teacher support and mental health. Based on prior research ([9]; [26]), we expected at least four domains of self-concept—scholastic, social competence, behavioral conduct, and global self-worth—would mediate the association between teacher support and mental health (Hypothesis 1). Second, we examined the area as a moderator in the association between teacher support and mental health. We hypothesized that area (rural vs. urban) would moderate the association between perceived teacher support and mental health (Hypothesis 2). Due to the lack of sufficient evidence, we do not set the direction of the moderating effect.

## 2. Method

### 2.1. Participants

In total, our sample consisted of 559 primary school children (44.54% girls) from two schools in China (one urban and one rural). These participants were enrolled in: grade 3 (*n* = 144; *M_a_*_ge_ = 8.90, *SD* = 0.05), grade 4 (*n* = 139; *M_a_*_ge_ = 9.94, *SD* = 0.05), grade 5 (*n* = 136; *M_a_*_ge_ = 10.85, *SD* = 0.49), and grade 6 (*n* = 140; *M_a_*_ge_ = 11.84, *SD* = 0.05).

The urban sample included 261 children (46.36% girls; *M_a_*_ge_ = 10.82, *SD* = 1.19) from a school located in Nanjing, a modern, economically developed megacity with nearly 10 million inhabitants. Nanjing has more educational and cultural resources for teaching and learning, as well as more opportunities for students to attend university than rural areas ([42]). The rural sample included 298 children (42.95% girls; *M*_age_ = 9.97, *SD* = 1.14) from a rural school located in a small village in Yangxin County, Hubei Province with about 1 million residents. The region is relatively traditional and economically underdeveloped, with a less balanced educational system.

### 2.2. Procedures

This study was conducted in accordance with the ethical principles of the Declaration of Helsinki. The Ethics Review Committee of University of Nanjing Normal University, to which the first author is affiliated, approved the data collection procedure. Proxy consent was obtained from school principals as well as homeroom teachers, who act in loco parentis in China when students are in school ([68]). Students were informed of their right to withdraw from the study at any time without any consequences.

We collected data in Nanjing in July 2021. Owing to scheduling challenges, the Yangxin data were collected later, in September 2021. On the day of data collection, students were invited to complete a questionnaire about perceptions about their teacher support, themselves, and mental health. Before students filled in the questionnaire, they received information from research assistants about this study and were instructed about its procedures. Students were informed that their responses would remain anonymous, unrelated to their scholastic grades, and would be used exclusively for scientific purposes. Research assistants were present throughout the process to answer students’ questions. After students completed the questionnaires, the assistants reviewed the questionnaires to ensure data completeness before collecting them. It took approximately 30 min for the students to complete the survey.

### 2.3. Instruments

#### 2.3.1. Teacher Support (Independent Variable)

We used the Perceived Teacher Support subscale from the Perceived School Climate Questionnaire ([11]) due to its established validity and reliability in Chinese samples ([35]). This subscale consists of 10 items (e.g., Teachers will help me when I am in trouble). Student rated these items on a 5-point Likert scale, ranging from 1 (never) to 5 (very often). Higher scores indicate that students perceive a higher level of teacher support. In the present study, internal consistency of the perceived teacher support subscale was 0.69 for the rural sample and 0.80 for the urban sample.

#### 2.3.2. Mental Health (Dependent Variable)

In this study, children’s mental health was assessed using two indicators: depression and anxiety. A composite mental health score was derived by averaging the scores of depression and anxiety. Higher scores on this composite indicate more depressive and anxious symptoms, signifying poorer mental health.

Depression was assessed with Depression Self-Rating Scale for Children, which measured the frequency of students experiencing depressive symptoms (DSRSC for short, [5]; Chinese version by [62]). The scale consists of 18 items (e.g., I sleep very well) which are rated on a 3-point Likert scale (0 = not at all, 1 = sometimes, 2 = often). Higher scores indicate more depressive symptoms. The DSRSC measure has been developed specifically for children and validated in the Chinese sample, demonstrating good validity and reliability ([5]; [62]). The internal consistency of the scale was 0.69 for the rural sample and 0.83 for the urban sample in the present study.

In addition to depression, anxiety was included as another indicator of children’s mental health. The Screen for Child Anxiety Related Emotional Disorders (SCARED for short; [4]; Chinese version by [72]) was used to assess children’s anxiety because it has been well validated and widely used in Chinese children ([34]; [72]). This questionnaire consists of 41 items which are rated on a 3-point Likert scale (0 = not at all, 1= sometimes, 2 = often). Five subscales were included: panic disorder/significant somatic symptoms (13 items), generalized anxiety disorder (9 items), separation anxiety (8 items), social anxiety disorder (7 items), and school avoidance (4 items). An example item is “When I feel frightened, it is hard to breathe”. Higher overall scores indicate more anxious symptoms. In the present study, the internal consistency for each subscale ranged from 0.51 to 0.75 for the rural sample, and from 0.70 to 0.87 for the urban sample.

#### 2.3.3. Self-Concept (Mediating Variable)

Students reported about their own self-concept with the Self-Perception Profile for Children (SPPC for short; [29], [31]; Chinese version revised by [22]). We chose SPPC, one of the most widely used scales for multidimensional self-concept ([18]), to assess children’s self-concept in six domains. This questionnaire consists of 36 items (6 items for each domain). Each item is presented as two opposite statements linked by “BUT”, with a dichotomous response option for each statement (“1 = really true for me, 2 = sort of true for me”). A sample sentence was presented in Figure 2. Six domains of self-concept was measured: scholastic competence (e.g., “Some kids feel that they are very good at their schoolwork BUT Other kids worry about whether they can do the school work assigned to them”), social competence (e.g., “Some kids find it hard to make friends BUT Other kids find it pretty easy to make friends”), athletic competence (e.g., “Some kids wish they could be a lot better at sports BUT Other kids feel they are good enough at sports”), physical appearance (e.g., “Some kids are happy with the way they look BUT Other kids are not happy with the way they look”), behavioral conduct (e.g., “Some kids often do not like the way they behave BUT Other kids usually like the way they behave”), and global self-worth (e.g., “Some kids are often unhappy with themselves BUT Other kids are pretty pleased with themselves”). This format creates a 4-point Likert scale, which has been suggested to reduce social desirability. Higher scores indicate a higher level of self-concept. Previous studies have supported the validity and reliability of this measurement ([22]; [31]). In the present study, the internal consistency of the scale ranged from 0.62 to 0.74 for the rural sample, and from 0.71 to 0.80 for the urban sample.

### 2.4. Data Analyses

Originally, our sample included 559 children. However, 38 children (6.80%) were excluded from the final data. Specifically, 29 students reported the same answers for all items or gave the answers with a certain pattern (5.19%), 8 students (1.43%) were absent on the survey day, and 1 student was reported by the homeroom teacher to have a psychiatric condition (0.18%). The final valid sample size was 525, including 264 rural children and 261 urban children. Missing data were handled using pairwise deletion to maximize the available data for each analysis ([33]). No outliers were detected using standardized scores (|z| > 3), indicating that all data points were within a reasonable range. Therefore, the assumptions of the analysis were met ([63]).

We conducted the analyses in SPSS 24.0. We selected Model 5 of the PROCESS macro to investigate our hypotheses, as this model enables the examination of direct effects, indirect effects, and interaction effects simultaneously. That is, this model allows us to account for mediation when testing moderation effects, and vice versa, thereby providing a more robust analysis ([32]). In this model, teacher support was entered as the independent variable, mental health as the dependent variable, domain-specific self-concept as the mediating variable, and area as the moderating variable. Conditional effects were further examined when a significant interaction effect was observed.

We tested the self-concept in six domains separately. Regarding mental health, to investigate whether the results may vary when the operationalization of mental health is different, we built separate models for depression, anxiety, and mental health (i.e., the mean of depression and anxiety). In total, we tested 18 models (6 domains × 3 variables of mental health). Concerning the potential effect of gender and age ([53]; [57]), we added them as covariates in each model.

## 3. Results

### 3.1. Descriptive Statistics

The correlations among the investigated variables are displayed in Table 1. Specifically, higher perceived teacher support was correlated with better mental health, i.e., lower depression and anxiety (*r*s = −0.31, −0.42, −0.22, *p*s < 0.01). Teacher support was positively associated with self-concept across each domain, including scholastic competence, social competence, athletic competence, physical appearance, behavioral conduct, and global self-worth (*r*s ranged from 0.15 to 0.36, *p*s < 0.01). Poorer mental health—higher depression and anxiety—were associated with lower self-concept across all six domains (*r*s ranged from −0.23 to −0.52, *p*s < 0.01). Compared to urban children, rural children reported poorer mental health, i.e., higher depression and anxiety (*r*s = 0.17, 0.28, 0.10, *p*s < 0.05), along with lower perceived teacher support (*r* = −0.51) and self-concept in five out of six domains (*r*s ranged from −0.26 to −0.11, *p*s < 0.05).

### 3.2. Relationships Between Teacher Support and Mental Health: The Mediating Role of Self-Concept and the Moderating Role of Area

As the results for depression, anxiety, and mental health (i.e., mean of depression and anxiety) were statistically consistent, we only present the results of mental health. Results for depression and anxiety can be found in the Appendix A (see Table A1, Table A2, Table A3 and Table A4). Results indicated that, after controlling for age and gender, perceived teacher support was associated with children’s mental health partially through self-concept in six domains: scholastic, social, and athletic competence, physical appearance, behavioral conduct, and global self-worth (supporting Hypothesis 1). Area significantly moderated the relationship between perceived teacher support and children’s mental health (supporting Hypothesis 2), such that the effect of perceived teacher support on mental health was more pronounced among urban children than rural children (see Figure 3).

The direct, indirect, and interaction effects of perceived teacher support on mental health are shown in Table 2. Model fitting index showed that all six models fitted the data well, with *R*^2^ ranged from 0.19 to 0.29, *F* ranged from 18.79 to 32.73, *p*s < 0.001. Regarding direct effects, higher perceived teacher support was associated with better mental health (direct effects ranged from −5.67 to −3.63, *p*s < 0.001). Regarding the mediating role of self-concept, 95% bootstrapped confidence intervals for all indirect effects did not contain zero. Therefore, indirect effects of perceived teacher support on mental health through each dimension of self-concept were significant (indirect effects ranged from −1.41 to −0.60). Children who perceived more teacher support tended to report higher self-concept, which in turn was associated with better mental health. Regarding the moderating role of area, in each model, the interaction effects of perceived teacher support and area on mental health were significant. Interaction effects ranged from 2.48 to 3.92, *p*s < 0.05 (see Table 2).

Conditional effects of areas were presented in Table 3. The negative association between perceived teacher support and mental health was stronger in urban than rural children (urban: effect ranged from −5.56 to −3.64, *p*s < 0.05; rural: effect ranged from −1.62 to −0.95, *p*s > 0.05 except for physical appearance).

## 4. Discussion

This study investigated the potential factors in the association between teacher support and mental health among Chinese children. Two main findings emerged from our analyses. First, self-concept across domains was found to mediate the relationship between perceived teacher support and children’s mental health. Second, the association between teacher support and mental health was moderated by area, with a significantly stronger association found in urban children than in rural children.

### 4.1. The Association Between Teacher Support and Mental Health and Its Mediating Role of Self-Concept

In line with our Hypothesis 1 ([7]; [58]; [60]), we found that teacher support was not only directly related to children’s mental health, but was also indirectly associated with mental health via the multidimensional self-concept. Specifically, children with higher perceived teacher support were more likely to have higher self-concept, which in turn contributed to better mental health. This finding provides support for the Thriving Through Relationships framework in the Chinese context ([26]). As one of the few studies that have been conducted in a non-Western setting (i.e., China), our findings extend existing work on the link between teacher support and mental health to an Eastern setting, highlighting the crucial role that teacher support plays in promoting children’s psychological well-being.

More importantly, our research indicates that the mediation effect was consistent across different domains of self-concept. Specifically, in line with prior research ([23]; [44]), we found that higher teacher support was associated with higher scholastic competence. Crucially, scholastic competence served as a significant mediator in the association between teacher support and mental health. It was not surprising that teacher support played an important role in promoting children’s scholastic competence, which was one main function of teachers in both Western and Eastern educational contexts ([13]; [19]; [23]). In turn, higher scholastic competence may further improve children’s mental health.

Notably, behavioral conduct mediated the association between teacher support and children’s mental health. This finding underlies another main function of teachers in the Chinese context: cultivating moral norms ([14]; [77]). In China, moral education is compulsory from grade 1 onward, earlier than in many Western countries ([15]; [49]). Despite the differences between educational systems, it seems that teachers in both contexts bear the responsibility for children’s behavioral regulation and exert an influence on their mental health cross-culturally ([24]; [47]).

Additionally, we found the mediation effects of self-concept in other domains as well, namely, athletic competence, physical appearance, and global self-worth. These findings suggest that teachers may help children develop a positive body image and satisfaction about themselves ([19]; [21]), further reinforcing the effect of teacher support on mental health. Taken together, our study underscores the important role of teacher support in fostering children’s mental health in both a direct and indirect way, demonstrating that psychological well-being is not only contingent on scholastic success but also extends to social, emotional, and physical domains.

### 4.2. The Moderating Role of Area in the Relationship Between Teacher Support and Mental Health

According to the Thriving Through Relationships framework, social support plays a significant role in children’s lives, even when they experience adversity or hardship ([26]). However, in reality, when considering the differences in specific contexts, this effect may be more complex. Regarding the area as a moderator, our findings revealed that area moderated the association between perceived teacher support and children’s mental health after controlling for the mediation effect of self-concept. In other words, teacher support showed a weaker association with mental health for rural children compared to urban children. One possible explanation lies in differences in the quantity and quality of teacher support available in rural versus urban areas ([41]; [74]). In our study, the average class size in rural school was nearly three times that in the urban schools. The large class size renders it more difficult for rural teachers to provide individualized support than urban teachers. Despite the fact that teachers in rural areas must manage excessive numbers of students, they have very limited access to professional support and development opportunities ([27]; [41]). This lack of training and resources might result in uneven teaching quality and increased professional burnout, further constraining the quality of support that teachers can provide to students. Therefore, the level of teacher support might fall short compared to the average in urban schools, which might weaken the protective effect of teacher support on mental health outcomes for rural children.

Another possible explanation lies in the fact that for rural children, family support may have a more pronounced impact on mental health than teacher support. For both rural and urban children, family support is crucial for their well-being ([9]; [25]; [75]). However, in rural areas, parents view education as solely the teachers’ responsibility and believe that children’s developmental outcomes are largely left to chance, with parents only needing to provide basic material necessities ([41]). As a result, children at home often struggle to obtain sufficient developmental resources and parental support, especially for those whose parents work away. In the absence of adequate family supervision and support, the protective role of teacher support in children’s mental health may appear negligible. The phrase “5 + 2 = 0” (referring to teachers’ efforts during the week being negated by the lack of parental supervision during the weekend) vividly captures rural teachers’ sense of helplessness stemming from children’s lack of adequate parental supervision ([27]). Rather than merely comparing and distinguishing the psychological differences between rural and urban children, it is more imperative to address the invisible disadvantages represented by rural contexts—such as lower family socioeconomic status, inadequate healthcare, and limited educational resources ([39])—as these may be more critical protective factors for child development.

Although this study did not aim to detect the prevalence of mental health disorder, it is noteworthy that our study found alarmingly high levels of depression and anxiety among both rural and urban children, exceeding the Chinese norm reported two decades ago ([62]; [72]; See Appendix B for more details). This finding suggests a concerning decrease in children’s mental health, which is consistent with a Chinese meta-analysis indicating a significant increase in depression levels among Chinese adolescents over the past three decades ([71]). This increase parallels China’s rapid economic growth, which has been accompanied by intensified scholastic, social, and healthcare pressures, widening socioeconomic disparities, unequal educational resource distribution, and a weakening sense of social connection, all of which may exert a detrimental influence on children’s mental health ([37]; [39]; [72]). Additionally, a recent study reported that the prevalence of depression and anxiety in Chinese adolescents was 25.9% and 46.6%, respectively ([73]), which is comparable to our findings. Overall, these findings indicate that the mental health status of contemporary Chinese children is concerning and warrants further attention.

### 4.3. Limitations

Five limitations should be considered when interpreting our findings. First, we only included one rural school and one urban school. This constrains the generalizability of our results across China. Nevertheless, this study provides a valuable reference for understanding children’s mental health across urban and rural settings. By examining the combined influences of regional and socio-psychological environments, our findings offer insights into the complex factors shaping children’s mental health. This approach is particularly relevant amid ongoing urbanization in developing countries, where the urban–rural divide remains significant ([38]; [39]; [76]). Future studies are suggested to include a larger and representative sample to see whether our findings can be generalized.

Second, the cross-sectional design does not permit conclusions about causal or temporal relations. While our findings suggest associations among teacher support, self-concept, and children’s mental health, the directionality of these associations remains unclear. It is possible that children with better mental health perceive higher teacher support. Alternatively, unexamined factors—such as limited socioeconomic opportunities in rural areas—may simultaneously contribute to both lower teacher support and poorer mental health. Future studies should adopt longitudinal and experimental designs to examine the stability and developmental trajectories of mental health, its association with teacher support over time, and to establish causal pathways ([6]; [20]).

Third, the reliance on self-report measures introduces the potential for bias, as children’s responses may be influenced by social desirability or momentary mood fluctuations, thereby potentially confounding the reported relationships. It would be valuable to incorporate multi-informant approach (e.g., teacher reports, observations) to disentangle the effect of perceived versus received social support on children’s self-concept and mental health.

Fourth, scales used in the present study demonstrated satisfactory reliability overall. However, in the rural sample, internal consistency reliability for the subscales assessing social competence and behavioral conduct, depression, and perceived teacher support, was slightly below the commonly accepted threshold of 0.70. One potential contributing factor may be differences in language exposure and development commonly observed between rural and urban populations ([79]), which may influence how well students understand the items included in the questionnaire. Notably, similar levels of reliability have been reported in some previous studies (e.g., [22]; [61]). Considering the brevity of these subscales (i.e., six items each), the observed reliability may still be regarded as acceptable. Further research is warranted to clarify these findings.

Lastly, this study did not examine household income or parental education. These socioeconomic factors are linked to children’s perceived teacher support ([1]; [2]), thereby possibly confounding the effects of area and teacher support on children’s mental health. Consequently, our results cannot distinguish whether the moderating role of area is shaped by socioeconomic factors (e.g., household income or parental education), or by other important variables (e.g., allocation and availability of supportive resources) derived from disparities between rural and urban areas. Therefore, it would be insightful for future research to replicate our findings while considering these socioeconomic factors.

### 4.4. Practical Implications

The alarmingly high rates of depression and anxiety among children in China ([25]; [36]), as also found in the present study, underscore an urgent need for effective interventions. Our findings on the relationship between teacher support and mental health offer practical insights for school teachers, mental health professionals, and policymakers to address these challenges.

For school teachers, it is important for them to foster children’s holistic and authentic self-concept by providing sufficient support. By offering support across multiple domains, teachers may help children recognize their diverse abilities and potential, thereby facilitating holistic self-evaluations and promoting children’s mental health ([12]; [45]). For example, in addition to acknowledging scholastic progress, teachers could offer positive feedback on athletic and social competence, while also providing emotional support for children with behavioral conduct ([47]). Additionally, encouraging students to engage in temporal comparison (i.e., measuring progress against their past performance rather than peers’ achievements) can help cultivate authentic self-esteem ([28]), a process that ultimately enhances children’s mental health.

For mental health professionals, they can support children’s psychological well-being through three ways. First, they can design and implement specialized courses and activities to directly enhance children’s self-concept and promote mental health. Second, they can collaborate closely with classroom teachers—particularly homeroom teachers—to deliver targeted, evidence-based Social-Emotional Learning (SEL) programs, equipping educators with skills to provide more effective support ([12]; [45]). Third, they can provide psycho-educational presentations or workshops for parents to help create nurturing home environments, which is especially critical for children in rural areas where parental supervision and parenting resources are often limited ([25]; [27]). Together, these efforts establish a comprehensive support system spanning school, family, and individual contexts.

For policymakers, given the disparities in teacher support and mental health outcomes between rural and urban children, it is important to invest in programs that promote either teacher support or children’s mental health in rural areas. Thus far, mental health teams in China’s primary and secondary rural schools remain underdeveloped. For instance, mental health classes are often given by non-specialists (e.g., subject teachers). Furthermore, both the curriculum and counseling services lack professional depth ([25]; [41]). Therefore, it is vital to strengthen policies that guarantee every rural school has at least one professional psychological counselor, fund targeted mental health training for teachers, and leverage community resources to support children’s well-being.

## Figures and Tables

**Figure 1 behavsci-15-00987-f001:**
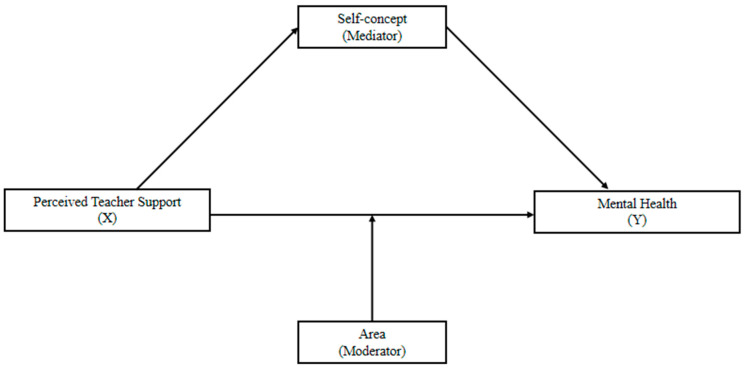
Conceptual model of study. Source: Authors.

**Figure 2 behavsci-15-00987-f002:**
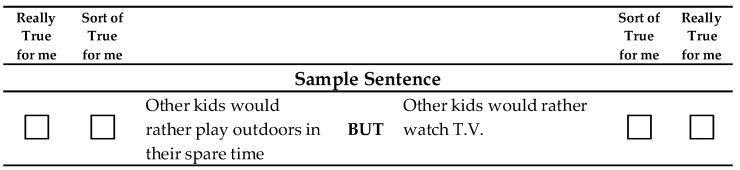
A sample sentence from the SPPC. Source: ([31]).

**Figure 3 behavsci-15-00987-f003:**
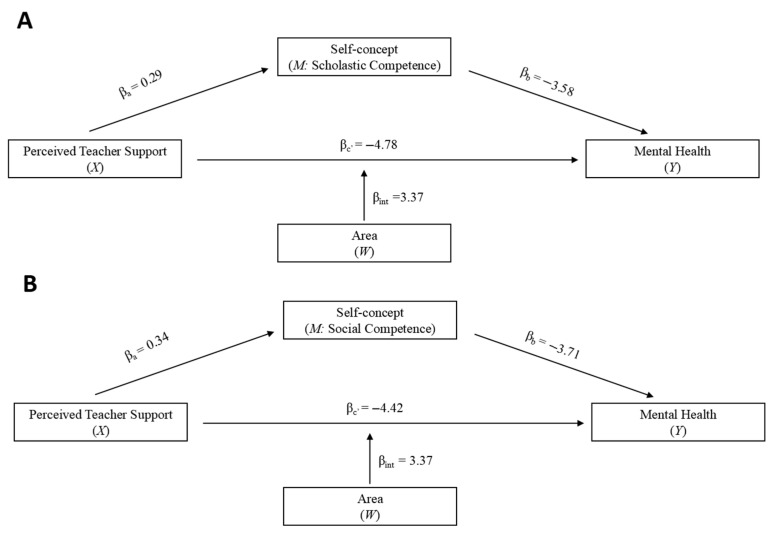
Relationship between perceived teacher support and mental health: Mediating role of self-concept and moderating role of area. Note. Six models were separately conducted across six domains of self-concept: (**A**) scholastic competence, (**B**) social competence, (**C**) athletic competence, (**D**) physical appearance, (**E**) behavioral conduct, and (**F**) global self-worth. int= X*W. All six models were conducted with age and gender as covariates. Source: Authors.

**Table 1 behavsci-15-00987-t001:** Descriptive statistics and correlations among included variables (*N* = 525).

	α	*M* (*SD*)	Skewness	Kurtosis	1	2	3	4	5	6	7	8	9	10	11	12
1 PTS	0.81	3.67 (0.76)	−0.20	−0.75	—											
2 MH	—	18.18 (9.00)	0.61	0.27	−0.31 **	—										
3 Depression	0.78	12.52 (5.67)	0.18	−0.28	−0.42 **	0.71 **	—									
4 Anxiety	0.93	23.83 (14.52)	0.71	0.29	−0.22 **	0.96 **	0.49 **	—								
5 Scholastic	0.75	2.53 (0.62)	0.22	−0.30	0.30 **	−0.33 **	−0.38 **	−0.25 **	—							
6 Social	0.76	2.70 (0.68)	−0.35	−0.38	0.36 **	−0.38 **	−0.47 **	−0.29 **	0.39 **	—						
7 Athletic	0.78	2.41 (0.71)	0.22	−0.60	0.15 **	−0.29 **	−0.34 **	−0.23 **	0.33 **	0.39 **	—					
8 Physical	0.76	2.58 (0.70)	−0.1	−0.62	0.19 **	−0.40 **	−0.41 **	−0.34 **	0.34 **	0.34 **	0.29 **	—				
9 Conduct	0.68	2.80 (0.55)	−0.16	−0.16	0.33 **	−0.34 **	−0.46 **	−0.24 **	0.52 **	0.46 **	0.26 **	0.40 **	—			
10 Self-worth	0.75	2.68 (0.63)	−0.21	−0.33	0.20 **	−0.45 **	−0.52 **	−0.35 **	0.49 **	0.47 **	0.27 **	0.65 **	0.57 **	—		
11 Area	—	0.50 (0.50)	−0.01	−2.01	−0.51 **	0.17 **	0.28 **	0.10 *	−0.15 **	−0.16 **	−0.16 **	−0.02	−0.11 *	−0.26 **	—	
12 Gender	—	0.54 (0.50)	−0.18	−1.98	−0.13 **	−0.11 *	−0.05	−0.12 **	0.04	0	0.12 **	0.08	−0.18 **	0.07	0.02	—
13 Age	—	10.37 (1.26)	0.12	−0.82	0.26 **	−0.09	0	−0.11 *	−0.07	0.03	−0.10 *	−0.17 **	−0.08	−0.19 **	−0.35 **	−0.03

*Note.* PTS = perceived teacher support; MH = mental health; Scholastic = scholastic competence; Social = social competence; Athletic = athletic competence; Physical = physical appearance; Conduct = behavioral conduct; Self-worth = global self-worth. Area: 0 = urban; 1 = rural. Gender: 0 = girls, 1 = boys; ** *p* < 0.01; * *p* < 0.05.

**Table 2 behavsci-15-00987-t002:** Direct, indirect, and interaction effects of perceived teacher support on mental health: Mediating role of self-concept and moderating role of area.

Effect	Path	*R*	*R* ^2^	*F*	β	*SE*	*t*	95%CI	
**(Scholastic competence)**									
Direct	PTS→MH	0.44	0.19	18.79 ***	−4.78	0.87	−5.52 ***	−6.48	−3.08
Indirect	PTS→Scholastic→MH				−1.04	0.25	-	−1.57	−0.57
Interaction	PTS × Area →MH				3.37	1.16	2.90 **	1.09	5.65
**(Social competence)**									
Direct	PTS→MH	0.45	0.20	20.36 ***	−4.42	0.87	−5.09 ***	−6.12	−2.71
Indirect	PTS→Social→MH				−1.27	0.27	-	−1.82	−0.76
Interaction	PTS × Area →MH				3.24	1.15	2.81 **	0.98	5.51
**(Athletic competence)**									
Direct	PTS→MH	0.44	0.19	19.03 ***	−5.67	0.84	−6.78 ***	−7.32	−4.03
Indirect	PTS→Athletic→MH				−0.60	0.18	-	−0.98	−0.29
Interaction	PTS × Area →MH				3.92	1.16	3.39 ***	1.64	6.19
**(Physical appearance)**									
Direct	PTS→MH	0.51	0.26	27.83 ***	−4.48	0.82	−5.47 ***	−6.09	−2.87
Indirect	PTS→Physical→MH				−1.16	0.25	-	−1.68	−0.70
Interaction	PTS × Area →MH				3.21	1.11	2.89 **	1.03	5.40
**(Behavioral conduct)**									
Direct	PTS→MH	0.46	0.21	20.83 ***	−4.37	0.87	−5.04 ***	−6.07	−2.67
Indirect	PTS→Conduct→MH				−1.22	0.27	-	−1.77	−0.73
Interaction	PTS × Area →MH				3.05	1.15	2.64 **	0.78	5.32
**(Global self-worth)**									
Direct	PTS→MH	0.54	0.29	32.73 ***	−3.63	0.82	−4.44 ***	−5.24	−2.03
Indirect	PTS→Worth→MH				−1.41	0.28	-	−1.99	−0.89
Interaction	PTS × Area →MH				2.48	1.09	2.27 *	0.33	4.62

*Note.* Separate models were built for each domain of self-concept to test its mediating role and the moderating role of area in the association between perceived teacher support and children’s mental health. PTS = perceived teacher support; MH = mental health; Scholastic = scholastic competence; Social = social competence; Athletic = athletic competence; Physical = physical appearance; Conduct = behavioral conduct; self-worth = global self-worth. *** *p* < 0.001, ** *p* < 0.01, * *p* < 0.05.

**Table 3 behavsci-15-00987-t003:** Conditional effects of area in relationships between perceived teacher support on mental health.

Path	Area	Effect	*SE*	*t*	*p*	LLCI	ULCI
(scholastic competence)	urban	−4.63	0.85	−5.43	<0.001	−6.31	−2.96
	rural	−1.25	0.82	−1.53	0.13	−2.85	0.35
(social competence)	urban	−4.17	0.86	−4.86	<0.001	−5.86	−2.49
	rural	−0.95	0.81	−1.17	0.24	−2.55	0.65
(athletic competence)	urban	−4.40	0.81	−5.44	<0.001	−5.99	−2.81
	rural	−1.22	0.78	−1.57	0.12	−2.75	0.30
(physical appearance)	urban	−5.56	0.83	−6.74	<0.001	−7.18	−3.94
	rural	−1.62	0.81	−2.01	0.04	−3.21	−0.04
(behavioral conduct)	urban	−4.34	0.87	−5.01	<0.001	−6.05	−2.64
	rural	−1.17	0.82	−1.43	0.15	−2.77	0.44
(global self-worth)	urban	−3.64	0.81	−4.52	<0.001	−5.22	−2.06
	rural	−1.15	0.76	−1.50	0.13	−2.64	0.35

## Data Availability

The original data are available from the first author upon request.

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
