# Peer review of "Taking a Closer Look at Teacher Support and Children’s Mental Health: The Mediating Role of Self-Concept and the Moderating Role of Area"

_behavsci, 2025, doi:10.3390/bs15070987_

Round 1
Reviewer 1 Report
Comments and Suggestions for Authors
Dear authors,
Your paper is very interesting, but there are some directions of improvement:
- Please add in Table 1 values of Cronbach's alpha for total sample
- Please emphasize in Method chapter independent and dependent variables
- Since your literature is a bit older, you could use this improvement in the Discussion part, that is a bit short: what about impact of household income and education of parents as potential moderator variables? Could you compare your results with some foreign research? Do you have any explanation for age and gender not being such important factors as area?
Author Response
|
Response to Reviewer 1 Comments
|
||
|
1. Summary |
|
|
|
Thank you very much for taking the time to review this manuscript. We appreciate the positive and constructive suggestions provided by the reviewer very much. Please find the detailed responses below and the corresponding revisions in highlight in the re-submitted files. Point-by-point responses to the reviewer are listed in respectively red and italics (quoted from the revised text) in this letter. |
||
|
2. Questions for General Evaluation |
Reviewer’s Evaluation |
Response and Revisions |
|
Is the content succinctly described and contextualized with respect to previous and present theoretical background and empirical research (if applicable) on the topic? |
Yes |
|
|
Are the research design, questions, hypotheses and methods clearly stated? |
Yes |
|
|
Are the arguments and discussion of findings coherent, balanced and compelling? |
Can be improved |
see Response 1, 2 and 3 (point-by-point) |
|
For empirical research, are the results clearly presented? |
Yes |
|
|
Is the article adequately referenced? |
Must be improved |
see Response 3 (point-by-point) |
|
Are the conclusions thoroughly supported by the results presented in the article or referenced in secondary literature? |
Can be improved |
see Response 3 (point-by-point) |
|
3. Point-by-point response to Comments and Suggestions for Authors |
||
|
Comments 1: Dear authors, Your paper is very interesting, but there are some directions of improvement: Please add in Table 1 values of Cronbach's alpha for total sample |
||
|
Response 1: We have added Cronbach's alpha for total sample to Table 1 in the Results section (pp.8, line. 346).
|
||
|
Comments 2: Please emphasize in Method chapter independent and dependent variables. |
||
|
Response 2: Thank you for your suggestion. To highlight independent and dependent variables in the Method section (pp. 6-7, line. 250-283), we have now (1) reordered the presentation of the instrument as follows: teacher support (independent variable), mental health (dependent variable), and self-concept (mediating variable). (2) clarified the roles of the key variables by explicitly indicating whether each is an independent, dependent, or mediating variable. Specifically, we added this information in the section titles of the instrument descriptions—for example, “Teacher support (independent variable)”. (3) retained the explanation when introducing the model we estimated as follow:
In The Data Analyses: “In this model, teacher support was entered as the independent variable, mental health as the dependent variable, domain-specific self-concept as the mediating variable, and area as the moderating variable. ”(pp. 8, line. 323-325)
|
||
|
Comments 3: Since your literature is a bit older, you could use this improvement in the Discussion part, that is a bit short: what about impact of household income and education of parents as potential moderator variables? Could you compare your results with some foreign research? Do you have any explanation for age and gender not being such important factors as area? |
||
|
Response 3: Thank you for these insightful questions for the Discussion.
(1) Regarding the impact of household income and parental education level: On the one hand, this study did not measure household income or parental education levels, therefore, we could not robustly demonstrate its impact based on our results. But we cautiously discussed this point when explaining the moderation effect of area.
In the Discussion section: “Rather than merely comparing and distinguishing the psychological differences between rural and urban children, it is more imperative to confront and clarify the invisible disadvantages represented by rural contexts—such as lower family economic status, inadequate healthcare, and limited educational resources (Li, 2016)—as these may be more critical protective factors for child development.”(pp. 12, line. 473-477)
On the other hand, these socioeconomic factors may moderate the relationship between teacher support and children's mental health (as you noted), which deserve more attention for future research. Therefore, we have now explicitly stated this limitation in our paper and discussed the significance of incorporating socioeconomic status in future research. Specific revisions can be found below:
In the Limitations section: “Lastly, although we controlled for gender and age in our model to account for their potential influence on the association between teacher support and mental health, we were not able to account for other potential factors such as household income or parental education. These socioeconomic factors may moderate the association between teacher support and mental health by shaping children’s responsiveness to teacher support (Atlay et al., 2019; Bakchich et al., 2023). For instance, children from lower-income families or with less-educated parents may have fewer emotional or academic resources at home, making teacher support more central to their well-being. In contrast, children from more advantaged backgrounds may receive substantial support outside of school, which could buffer or dilute the impact of teacher support on their mental health. As such, the strength of the association between teacher support and mental health may vary across different socioeconomic groups. Therefore, it would be insightful for future research to replicate our findings while taking these socioeconomic factors (e.g., household income, parental education) into consideration.” (pp. 13-14, line. 532-540)
References added: Atlay, C., Tieben, N., Fauth, B., & Hillmert, S. (2019). The role of socioeconomic background and prior achievement for students’ perception of teacher support. British Journal of Sociology of Education, 40(7), 970–991. https://doi.org/10.1080/01425692.2019.1642737 Bakchich, J., Carré, A., Claes, N., & Smeding, A. (2023). The moderating role of socioeconomic status on the relationship between teacher social support and sense of belonging to school. British Journal of Educational Psychology, 93(1), 153-166. http://doi.org/10.1111/bjep.12545
(2) Regarding comparing our results with foreign research: To our best knowledge, there is no research exploring the mediating role of self-concept as well as the moderating role of area in the association between teacher support and mental health. Therefore, in the Discussion section, we confirmed that bivariate correlations (i.e., among teacher support, mental health, and self-concept) align with Western research. For instance, “Specifically, in line with prior research (Eccles & Roeser, 2006; Ma et al., 2021), we found that higher teacher support was associated with higher scholastic competence. Crucially, scholastic competence served as a significant mediator in the association between teacher support and mental health. ”(pp. 11, line. 416-419)
Furthermore, in order to provide a more nuanced comparison between our findings and the broader literature, we have now elaborated more on similarities and the differences between the Chinese context and the Western context in discussing associations between teacher support and children’s self-concept. Specific revisions are below.
In the Discussion section: “....... It was not surprising that teacher support played an important role in promoting children’s scholastic competence, which was one main function of teachers in both Western and Eastern educational contexts (Chen, 2005; Davidson et al., 2010; Eccles & Roeser, 2006). In turn, higher scholastic competence may further improve children’s mental health.
Notably, behavioral conduct mediated the association between teacher support and children’s mental health. This finding underlies another main function of teachers in the Chinese context: cultivating moral norms (Chen et al., 2004; Yau et al., 2009). In China, moral education is compulsory from Grade 1 onward, earlier than in many Western countries (Cheng et al., 2021; OECD, 2019). Despite the differences between educational systems, it seems that teachers in both contexts bear the responsibility for children’s behavioral regulation and exert an influence on their mental health cross-culturally (Ettekal & Shi, 2020; Merritt et al., 2012). ”(pp. 11, line. 419-431)
References added: OECD (2019), Education at a Glance 2019: OECD Indicators, OECD Publishing, Paris, https://doi.org/10.1787/f8d7880d-en.
(3) Regarding the explanation for age and gender not being such important factors as area: We fully agreed that age and gender may be significant factors in the relationship between teacher support and mental health, which was also partially reflected in the correlational analyses in Table 1. However, as these factors were not the primary focus of the present study, we included both variables as covariates in the model to statistically control for their potential influence.
In the Data Analyses section: “Concerning the potential effect of gender and age (Orth et al., 2014; Rueger et al., 2014), we added them as covariates in each model.”(pp. 8, line. 331-332)
In addition, this study specifically focuses on the role of area (urban-rural context) in moderating teacher support and mental health outcomes. This focus is motivated by geographic disparities in children’s mental health remain understudied. Such evidence is urgently needed in developing countries, where resource disparities between areas are substantial (Li, 2016).
|
||
|
4. Response to Comments on the Quality of English Language |
||
|
Point 1: The English is fine and does not require any improvement. |
||
|
Response 1: Thanks for your evaluation.
|
||
|
5. Additional clarifications |
||
|
To ensure the overall quality of the manuscript, we carefully proofread the entire manuscript. In doing so, we corrected typos and revised long or unclear sentences to improve clarity and readability. All specific changes can be found in the manuscript. |
||
Reviewer 2 Report
Comments and Suggestions for Authors
Thank you for inviting me to review this interesting study which tries to look at existing knowledge in a Chinese context. There is a lot of data and overall the paper certainly has merit for publication.
However, I have struggled with the presentation of results and discussion and I think a major reworking is needed. This relates to the way findings are presented as having negative correlations between variables. An example of this can be found in the discussion:
"Specifically, perceived teacher support was negatively associated with both children’s depression and anxiety. " l.390
I cannot tell if this means as teacher support is perceived as good, anxiety and depression are worse or if it means as teacher support is perceived as good anxiety and depression are better. Obviously this is totally critical to the understanding of the results. I would suggest that the authors remove all reference to negative correlations and spell out exactly what is meant in accessible language. For this reason I have given a low score for logical coherence but this is easily rectified by taking a step back from the stats and spelling the findings out clearly. It occurs throughout the paper
Specific comments:
- Please do a careful proof read. There are a number of places with missing words and typos e.g.l47 delete almost.
- l199-203 rewrite
- l250 Self concept, perhaps include an example of a question as the layout of the questionnaire is unusual.
- Please explain why you chose the questionairres. You describe them, but why you chose them in preference to others.
Author Response
|
Response to Reviewer 2 Comments
|
||
|
1. Summary |
|
|
|
Thank you very much for taking the time to review this manuscript. We appreciate the positive and constructive suggestions provided by the reviewer very much. Please find the detailed responses below and the corresponding revisions in highlight in the re-submitted files. Point-by-point responses to the reviewer are listed in respectively red and italics (quoted from the revised text) in this letter. |
||
|
2. Questions for General Evaluation |
Reviewer’s Evaluation |
Response and Revisions |
|
Is the content succinctly described and contextualized with respect to previous and present theoretical background and empirical research (if applicable) on the topic? |
Yes |
|
|
Are the research design, questions, hypotheses and methods clearly stated? |
Can be improved |
|
|
Are the arguments and discussion of findings coherent, balanced and compelling? |
Must be improved |
see Response 1, 2, 3, 4, and 5 (point-by point) |
|
For empirical research, are the results clearly presented? |
Can be improved |
see Response 1 (point-by point) |
|
Is the article adequately referenced? |
Yes |
|
|
Are the conclusions thoroughly supported by the results presented in the article or referenced in secondary literature? |
Can be improved |
see Response 1 (point-by point) |
|
3. Point-by-point response to Comments and Suggestions for Authors |
||
|
Comments 1: Thank you for inviting me to review this interesting study which tries to look at existing knowledge in a Chinese context. There is a lot of data and overall the paper certainly has merit for publication. However, I have struggled with the presentation of results and discussion and I think a major reworking is needed. This relates to the way findings are presented as having negative correlations between variables. An example of this can be found in the discussion: "Specifically, perceived teacher support was negatively associated with both children’s depression and anxiety. " l.390 I cannot tell if this means as teacher support is perceived as good, anxiety and depression are worse or if it means as teacher support is perceived as good anxiety and depression are better. Obviously this is totally critical to the understanding of the results. I would suggest that the authors remove all reference to negative correlations and spell out exactly what is meant in accessible language. For this reason I have given a low score for logical coherence but this is easily rectified by taking a step back from the stats and spelling the findings out clearly. It occurs throughout the paper |
||
|
Response 1: Thank you for your feedback. We found this suggestion really helpful. Following your suggestion, we have removed references to 'negative correlations' and instead used more descriptive language to present our results. We provide details of our revisions below.
1. In the Methods section, we have clearly stated that the mental health composite score. That is, higher scores indicate more depressive and anxious symptoms, signifying poorer mental health.
“A composite mental health score was derived by averaging the scores of depression and anxiety. Higher scores on this composite indicate more depressive and anxious symptoms, signifying poorer mental health. “(pp. 6, line. 260-262)
2. We have replaced phrasing such as 'X was negatively associated with Y' with more descriptive language, for example: 'higher X was associated with lower Y'.
In the Results section: “3.1. Descriptive Statistics ...... Specifically, higher perceived teacher support was correlated with better mental health—lower levels of depression and anxiety (rs = -.31, -.42, -.22, ps < .01). ......Poorer mental health — higher depression and anxiety — were associated with lower self-concept across all six domains (rs ranged from -.23 to -.52, ps < .01). Compared to urban children, rural children reported poorer mental health — higher depression and anxiety (rs = .17, .28, .10, ps < .05), along with lower perceived teacher support (r = -.51) and self-concept in five out of six domains (rs ranged from -.26 to -.11, ps < .05) . (pp. 8, line. 336-343)
In the Discussion section: "Specifically, children with higher perceived teacher support were more likely to have higher self-concept, which in turn contributed to better mental health. "(pp. 11, line. 408-409)
3. We have added some explanations for the statistical descriptions. In the Results section: “3.2. Relationships Between Teacher Support and Mental Health: The Mediating Role of Self-Concept and the Moderating Role of Area Therefore, indirect effects of perceived teacher support on mental health through each dimensions of self-concept were significant (indirect effects ranged from -1.41 to -.60). Children who perceived more teacher support were more likely to report higher self-concept, which in turn contributed to better mental health.“ (pp.8 , line.369-372)
4. When necessary, we retained the description of “negative association” to keep concise and clear. In the Results section: “The negative association between perceived teacher support and mental health was stronger in urban than rural children (urban: effect ranged from -5.56 to -3.64, ps <.05; rural: effect ranged from -1.62 to -0.95, ps > .05 except for physical appearance). “ (pp.10, line.382-384)
|
||
|
Comments 2: Please do a careful proof read. There are a number of places with missing words and typos e.g.l47 delete almost. |
||
|
Response 2: Thank you for your reminder. We have deleted the word “almost” as suggested(pp. 2, line. 48). To ensure that all typos and inconsistencies were corrected, we carefully proofread the manuscript, focusing on the following aspects: (1) Deleted redundant blanks by finding two more consecutive blanks in the whole paper one by one; (2) Corrected typos through careful word-by-word reading—for example, corrected “suport” into “support”; (3) Rewrote unclear or lengthy sentences to improve conciseness and clarity—An example can be seen in our next response to comment 3 (point-by-point response) ; (4) Replace informal expression with more formal ones—for example, changing “all together” into “in total”; (5) Ensured consistent verb tense throughout the manuscript, in accordance with APA 7th edition guidelines—for example, using past tense or perfect tense in the introduction when introducing previous work; (6) Standardized terminology—for example, replaced the interchangeable use of “scholastic competence” and “academic competence” with the consistent use of “scholastic competence”;
Given the extensive modifications made, we will not list each detail here. Specific revisions can be found in the resubmitted file.
|
||
|
Comments 3: l199-203 rewrite. |
||
|
Response 3: We have rewritten the sentence line.199-203 to better clarify our research contribution. Changes can be found below.
In The Present Study section: “Second, these findings would demonstrate how teacher support related to children’s mental health by identifying multidimensional self-concept as a mediator and area as a moderator, thus contributing to the Thriving Through Relationships framework.” (pp.5, line. 199-202)
|
||
|
Comments 4: l250 Self concept, perhaps include an example of a question as the layout of the questionnaire is unusual. |
||
|
Response 4: We agree. Therefore, we have now provided a sample sentence in the Instruments section as follow.
“A sample sentence was presented in Figure 2.” (pp. 7, line. 290, 306-307)
Figure 2. A sample sentence from the SPPC. Source: Harter, 2012.
|
||
|
Comments 5: Please explain why you chose the questionairres. You describe them, but why you chose them in preference to others. |
||
|
Response 5: Thanks for offering us opportunity to clarify this point. The scales were selected primarily based on two considerations: First, they are all child-specific scales appropriate for the developmental level of the participants in this study; second, they are well-established measures with demonstrated validity and reliability, both in their original versions and in studies conducted with Chinese samples. To clarify this, we have provided this information for each questionnaire we used in the Instruments section, as outlined below.
(1) For the Perceived Teacher Support subscale: “We used the Perceived Teacher Support subscale from the Perceived School Climate Questionnaire (Chen & Li, 2009) due to its established validity and reliability in Chinese samples (Li et al., 2013). ”(pp. 6, line.251-253)
(2) For the Depression Self-Rating Scale for Children (DSRSC for short): “The DSRSC measure has been developed specifically for children and validated in the Chinese sample, demonstrating good validity and reliability (Birleson et al., 1987; Su et al., 2003). “ (pp. 6-7, line. 267-269)
(3) For the Screen for Child Anxiety Related Emotional Disorders (SCARED for short): “The Screen for Child Anxiety Related Emotional Disorders (SCARED for short; Birmaher et al., 1997; Chinese version by Wang et al., 2002) was used to assess children’s anxiety because it has been well validated and widely used in Chinese children (Lei et al., 2020; Wang et al., 2002). “(pp.7, line. 272-274)
Reference added: Lei, X., Li, S., Wang, Y., Yang, Y., & Yang, Y. (2020). Analysis of Influencing Factors for Anxiety-related Emotional Disorders in Children and Adolescents During Home Quarantine in the COVID-19 Pandemic. Chinese Journal of Child Health Care, 28(4), 407-410. http://doi.org/10.11852/zgetbjzz2020-0169
(4) For Self-Perception Profile for Children (SPPC for short): “We chose SPPC, one of the most widely used scales for multidimensional self-concept (Dapp & Orth, 2024), to assess children’s self-concept in six domains.......This format creates a 4-point Likert scale, which has been suggested to reduce social desirability. Higher scores indicate a higher level of self-concept. Previous studies have supported the validity and reliability of this measurement (Ding et al., 2014; Harter, 2012).“(pp. 7, line. 286-287, 300-303)
Reference added: Dapp, L. C., & Orth, U. (2024). Rank-order stability of domain-specific self-esteem: A meta-analysis. Journal of Personality and Social Psychology, 127(2), 432-454. http://doi.org/10.1037/pspp0000497
|
||
|
4. Response to Comments on the Quality of English Language |
||
|
Point 1: The English could be improved to more clearly express the research. |
||
|
Response 1: Thanks for your feedback. Please see our response to Comment 2 (point-by-point response) above for specific language improvements we have made. |
||
Reviewer 3 Report
Comments and Suggestions for Authors
Dear authors,
Thank you for trusting me to review your paper entitled "Taking a Closer Look at Teacher Support and Children’s Mental Health: The Mediating Role of Self-Concept and the Moderating Role of Area". I firmly believe that your strudy’s findings offer meaningful insights into the multidimensional roles of teacher support and self-concept in influencing mental health outcomes.
Only two minor issues need to be solved in my opinion.
Firstly, the internal consistency of the instruments used in this study is generally satisfactory, but there are some concerns depending on the values below the standard treshold of 0.70 which may indicate weaker reliability. These values might still be tolerable in exploratory studies, but you should acknowledge this limitation. I recommend that you should discuss these discrepancies in your manuscript, particularly for subscales with reliability below 0.70.
Secondly -but this is only for helping readers assess the strengh of the findings- I would like to see a more specific Figure indicating schematically your results.
I have to congratulate you for your work.
Kindest regards
Author Response
|
Response to Reviewer 3 Comments
|
||
|
1. Summary |
|
|
|
Thank you very much for taking the time to review this manuscript. We appreciate the positive and constructive suggestions provided by the reviewer very much. Please find the detailed responses below and the corresponding revisions in highlight in the re-submitted files. Point-by-point responses to the reviewer are listed in respectively red and italics (quoted from the revised text) in this letter. |
||
|
2. Questions for General Evaluation |
Reviewer’s Evaluation |
Response and Revisions |
|
Is the content succinctly described and contextualized with respect to previous and present theoretical background and empirical research (if applicable) on the topic? |
Yes |
|
|
Are the research design, questions, hypotheses and methods clearly stated? |
Yes |
|
|
Are the arguments and discussion of findings coherent, balanced and compelling? |
Must be improved |
see Response 1 (point-by point) |
|
For empirical research, are the results clearly presented? |
Must be improved |
see Response 2 (point-by point) |
|
Is the article adequately referenced? |
Yes |
|
|
Are the conclusions thoroughly supported by the results presented in the article or referenced in secondary literature? |
Yes |
|
|
3. Point-by-point response to Comments and Suggestions for Authors |
||
|
Comments 1: Thank you for trusting me to review your paper entitled "Taking a Closer Look at Teacher Support and Children’s Mental Health: The Mediating Role of Self-Concept and the Moderating Role of Area". I firmly believe that your study’s findings offer meaningful insights into the multidimensional roles of teacher support and self-concept in influencing mental health outcomes. Only two minor issues need to be solved in my opinion. Firstly, the internal consistency of the instruments used in this study is generally satisfactory, but there are some concerns depending on the values below the standard treshold of 0.70 which may indicate weaker reliability. These values might still be tolerable in exploratory studies, but you should acknowledge this limitation. I recommend that you should discuss these discrepancies in your manuscript, particularly for subscales with reliability below 0.70. |
||
|
Response 1: Thank you for pointing this out. The reliabilities of our scale range from sufficient to satisfactory overall. However, the reliability coefficients for social competence and conduct behavior, depression and perceived teacher support in rural children fall below the commonly accepted threshold of .70, which warrants further attention. To address this, we now reflect on the relatively low reliability of these (sub-)scales in the Limitations section as follow:
“Fourth, scales used in the present study demonstrated satisfactory reliability overall. However, in the rural sample, internal consistency reliability for the subscales assessing social competence and conduct behavior, depression and perceived teacher support, was slightly below the commonly accepted threshold of .70. One potential contributing factor may be differences in language exposure and development commonly observed between rural and urban populations (Zheng et al., 2022), which may influence how well students understand the items included in the questionnaire. Notably, similar levels of reliability have been reported in some previous studies (e.g., Ding et al., 2013; Stewart et al., 2010). Considering the brevity of these subscales (i.e., six items each), the observed reliability may still be regarded as acceptable. Further research is warranted to clarify these findings.”(pp.13, line.521-531)
References added: Stewart, P. K., Roberts, M. C., & Kim, K. L. (2010). The Psychometric Properties of the Harter Self-Perception Profile for Children with At-Risk African American Females. Journal of Child and Family Studies, 19(3), 326-333. http://doi.org/10.1007/s10826-009-9302-x Zheng, Y., Li, D., Chen, Z., & Liu, G. (2022). Picture book reading on the development of preschoolers in rural areas of China: Effects on language, inhibition, and theory of mind. Frontiers in Psychology, 13, 1030520. http://doi.org/10.3389/fpsyg.2022.1030520 |
||
|
Comments 2: Secondly -but this is only for helping readers assess the strengh of the findings- I would like to see a more specific Figure indicating schematically your results. |
||
|
Response 2: Following your suggestion, we have now presented our results in Figure 3 (A-F) (pp. 10-11, line. 388-395). We hope this facilitates a clearer understanding of our findings. |
||
|
4. Response to Comments on the Quality of English Language |
||
|
Point 1: The English is fine and does not require any improvement. |
||
|
Response 1: Thank you for your evaluation. |
||
|
5. Additional clarifications |
||
|
To ensure the overall quality of the manuscript, we carefully proofread the entire manuscript. In doing so, we corrected typos and revised long or unclear sentences to improve clarity and readability. All specific changes can be found in the manuscript. |
||
Round 2
Reviewer 2 Report
Comments and Suggestions for Authors
The authors have addressed all the points in my initial review and I think this has greatly improved the paper and made the findings clearer.
I am happy with the changes and think the paper is ready for publicaion.